# Oral Intake of L-Ornithine-L-Aspartate Is Associated with Distinct Microbiome and Metabolome Changes in Cirrhosis

**DOI:** 10.3390/nu14040748

**Published:** 2022-02-10

**Authors:** Angela Horvath, Julia Traub, Benard Aliwa, Benjamin Bourgeois, Tobias Madl, Vanessa Stadlbauer

**Affiliations:** 1Department of Gastroenterology and Hepatology, Medical University of Graz, 8036 Graz, Austria; angela.horvath@medunigraz.at (A.H.); benard.aliwa@medunigraz.at (B.A.); 2Center of Biomarker Research in Medicine, CBmed GmbH Graz, 8010 Graz, Austria; 3Department of Clinical Medical Nutrition, University Hospital Graz, 8036 Graz, Austria; Julia.Traub@uniklinikum.kages.at; 4Gottfried Schatz Research Center for Cell Signaling, Metabolism and Aging Molecular Biology and Biochemistry, Medical University of Graz, 8036 Graz, Austria; benjamin.bourgeois@medunigraz.at (B.B.); tobias.madl@medunigraz.at (T.M.); 5BioTechMed-Graz, 8010 Graz, Austria

**Keywords:** microbiome, metabolome, L-ornithine-L-aspartate, cirrhosis, hepatic encephalopathy

## Abstract

L-ornithine L-aspartate (LOLA) is administered as a therapeutic and/or preventive strategy against hepatic encephalopathy either intravenously or orally in patients with liver cirrhosis. Here, we analyzed how LOLA influences the microbiome and metabolome of patients with liver cirrhosis. We retrospectively analyzed the stool microbiome, stool, urine and serum metabolome as well as markers for gut permeability, inflammation and muscle metabolism of 15 cirrhosis patients treated orally with LOLA for at least one month and 15 propensity-score-matched cirrhosis patients without LOLA. Results were validated by comparing the LOLA-treated patients to a second set of controls. Patients with and without LOLA were comparable in age, sex, etiology and severity of cirrhosis as well as PPI and laxative use. In the microbiome, *Flavonifractor* and *Oscillospira* were more abundant in patients treated with LOLA compared to the control group, while alpha and beta diversity were comparable between groups. Differences in stool and serum metabolomes reflected the pathophysiology of hepatic encephalopathy and confirmed LOLA intake. In the urine metabolome, ethanol to acetic acid ratio was lower in patients treated with LOLA compared to controls. LOLA-treated patients also showed lower serum levels of insulin-like growth factor (IGF) 1 than patients without LOLA. No differences in gut permeability or inflammation markers were found. A higher abundance of *Flavonifractor* and *Oscillospira* in LOLA-treated patients could indicate LOLA as a potential microbiome modulating strategy in patients with liver disease. The lower levels of IGF1 in patients treated with LOLA suggest a possible link between the pathophysiology of hepatic encephalopathy and muscle health.

## 1. Introduction

Chronic liver diseases and liver cirrhosis are on the rise in Europe [1]. Hepatic encephalopathy (HE) is one of the most debilitating complications of cirrhosis and is associated with increased morbidity and mortality [2]. Although the pathogenesis is still incompletely understood, currently, it is assumed that protein and urea breakdown by colonic bacteria leads to ammonia release, and due to the reduced capacity of the liver to detoxify, ammonia accumulates and is shunted into the systemic circulation [3,4]. Ammonia accumulates in the brain and has neurotoxic effects, especially on astrocytes [5,6]. Available treatment modalities to lower ammonia include L-ornithine-L-aspartate (LOLA), nonabsorbable disaccharides (lactulose) and nonabsorbable antibiotics (rifaximin) [7]. The amino acids L-ornithine and L-aspartate in LOLA dissociate readily and are consecutively absorbed.

L-ornithine serves as an intermediary in the urea cycle in periportal hepatocytes in the liver and as an activator of carbamoyl phosphate synthetase, and both amino acids also lower ammonia levels by transamination to glutamate via glutamine synthetase in perivenous hepatocytes as well as by skeletal muscle and the brain [8]. Besides these direct effects, additional functions such as hepatoprotective effects or effects on restoring skeletal muscle proteostasis have been proposed but are not fully explained to date [9]. Since oral LOLA comes into contact with the human gut microbiome and the intestinal barrier, effects on the gut microbiome and intestinal permeability may play a role in the clinical effects of LOLA. The gut microbiome is not only involved in ammonia generation and thereby in the pathogenesis of HE, but it is today also understood as a key driver of complications of cirrhosis [10]. The gut microbiome is severely altered in liver cirrhosis in general, with a reduction in bacterial diversity and an increase in potential pathogens [11]. Factors influencing the microbiome in cirrhosis are etiology and severity of liver disease, drug intake, nutritional status, and inflammation [12]. In general, many human-targeted drugs can alter the composition of the gut microbiome [13]. For LOLA thus far, no data are available on its association with gut microbiome composition and metabolomic changes in cirrhosis. Therefore, we aimed to investigate whether LOLA intake in patients with liver cirrhosis is associated with taxonomic and functional changes of the gut microbiome and the urinary, and/or serum and/or fecal metabolome.

## 2. Materials and Methods

From an ongoing cohort study (NCT03080129, 29-280 ex 16/17), 15 patients with cirrhosis with oral LOLA intake for at least 1 month and 15 control cirrhotic patients were selected with nearest-neighbor propensity score matching. Propensity score matching was based on cirrhosis severity (Child-Pugh score and MELD-score), using the R package “MatchIt” [14,15]. To avoid overfitting and false positive results, results were validated by comparing the patients receiving LOLA to a second control group. This control group was selected by extending the regression model for nearest neighbor propensity score matching to include etiology of cirrhosis and proton pump inhibitor use in addition to disease severity. Due to this alteration, 9 out of 15 control patients (60%) were replaced. Sarcopenia is a potential confounder in the analysis too. To approximate its influence, LOLA patients were propensity score matched to a third control group of patients not receiving LOLA based on their sarcopenia status (non-sarcopenic, pre-sarcopenic, sarcopenic). To minimize confounding by potentially unbalanced parameters in regard to liver disease, this group comparison was only performed for sarcopenia related biomarkers with Mann–Whitney U tests.

Then, 16S rDNA sequencing data (for details regarding the method, see Appendix A) of these 30 patients were preprocessed with the QIIME 2 pipeline on a local Galaxy instance (https://galaxy.medunigraz.at, accessed on 21 September 2021) [16] and analyzed on the web-based platform “Calypso” V8.84 (https://cgenome.net, accessed on 25 September 2021) [17]. Taxonomic analysis was performed with a Bayesian classifier based on the SILVA V132 database. Low abundance filtering was applied to remove features that were present in only one sample or had less than 10 copies in total; cyanobacteria and chloroplasts were removed as likely contaminants. For alpha diversity approximations, a rarefied feature table with 8427 reads per sample was used to calculate richness, evenness, Chao-1 index and Shannon index. For beta diversity analysis and taxon comparisons, a full (not rarefied) feature table was subjected to Hellinger transformation (i.e., total sum scaling and square root transformation). Principal coordinate analysis (PCoA) with subsequent analysis of similarities (ANOSIM) based on Bray–Curtis dissimilarity as well as redundancy analysis (RDA) and non-parametric multidimensional scaling (NMDS) was used to identify similarities of microbiome structures between groups. LDA effect size (LEfSe) and analysis of compositions of microbiomes (ANCOM) identified differentially abundant taxa between the groups.

Nuclear magnetic resonance (NMR) metabolomic profiling was performed for urine, serum and stool samples using a well-established pipeline (for details regarding the method, see Appendix A) [18,19]. Metabolites were extracted using methanol, NMR spectra were recorded and processed in Matlab to obtain aligned and normalized datasets. Valid datasets were obtained from 29 patients (14 LOLA, 15 noLOLA), 27 patients (13 LOLA, 14 noLOLA) and 27 patients (13 LOLA, 14 noLOLA). The human metabolome database (HMDB) was used for metabolite annotation. Obtained concentrations were then normalized to the sum of all concentrations in a sample, square root transformed, mean centered and divided by the standard deviation of the respective feature for statistical analysis. Principal component analysis (PCA) was performed to visually assess the similarity of metabolome profiles between groups. To further discriminate groups, biomarker identification was performed based on area under the receiver operator characteristics curve (AUROC). For biomarker identification, the 20 ratios of metabolites with the lowest *p* value in group comparison tests were included in the list of analytes. Analysis was performed in the web-based version of MetaboAnalyst 5.0 (https://www.metaboanalyst.ca, accessed on 28 September 2021) [20].

Targeted metabolomics were compared between groups, whereby routine and experimental parameters were selected to complement the NMR results. Parameters include routine biomarkers for liver function/injury and related information: alanine aminotransferase (ALT), aspartate aminotransferase (AST), alkaline phosphatase (AP), gamma-glutamyltransferase (GGT), albumin, bilirubin, prothrombin time international normalized ratio (PZINR), C-reactive protein and total protein biomarkers for inflammation and gut permeability: fecal calprotectin, fecal zonulin, diamine oxidase (DAO), LPS binding protein and sCD14 indicators for sarcopenia: fibroblast growth factor (FGF) 21, irisin, myostatin and insulin-like growth factor (IGF)—1 biomarkers for neutrophil function: resting burst, priming and ROS production after *E. coli* stimulation, all given as percentage of positive cells and geometric mean of fluorescence intensity (GMFI) [20].

Biomarkers for liver function and injury as well as C-reactive protein were taken from the routine biochemistry report of the patients, fecal zonulin and fecal calprotectin were measured externally by Biovis’ Diagnostik MVZ GmbH (Limburg–Offenheim, Germany); neutrophil function was assessed by flow cytometry in heparinized whole blood using Phagoburst kits (Celonic, Basel, Switzerland) in house (for details, see Appendix A), the remaining biomarkers were assessed in house by ELISA (diamine oxidase and myostatin: Immundiagnostik, Bensheim, Germany; insulin-like growth factor-1 and fibroblast growth factor-21: Biotechne, Minneapolis, MN, USA; Irisin: Biovendor, Brno, Czeck Republic; LPS binding protein: Hycult Biotech, Uden, The Netherlands; sCD14: R&D systems, Minneapolis, MN, USA). All tests were performed according to manufacturers’ instructions. In addition, muscle function and muscle mass assessments (chair rise test, gait speed, midarm muscle circumference, hand grip strength and body mass index) were included in the analysis.

All analyses and visualizations, if not otherwise stated in the Method section, were performed with R (version 4.0.3) in R studio (version 1.4.1103) using the packages “tidyverse”, “readxl”, “writexl”, and “ggpubr” [14,21,22,23,24].

## 3. Results

### 3.1. Patient Characteristics

Of the 156 patients with a valid sarcopenia diagnosis and microbiome profile, 98 were diagnosed with cirrhosis. Of these 98, 95 had a microbiome profile with at least 8000 reads per sample, and 15 had a documented oral LOLA-therapy at the timepoint of sampling (for at least one month, range: 1–29 months). Out of the remaining 80 eligible patients without LOLA therapy, 15 patients were selected as a control group with nearest-neighbor propensity score matching, as shown in Figure 1. The two resulting groups (LOLA and noLOLA) were comparable in etiology of cirrhosis (*p* = 0.7), Child-Pugh score (*p* < 0.99), MELD-score (*p* = 0.5), age (*p* = 0.9), sex (*p* = 0.7), proton pump inhibitor use (*p* > 0.99) and lactulose use (*p* = 0.7). Non-absorbable antibiotics (rifaximin) were used by four patients in the LOLA group, but none in the control group (*p* = 0.1). For patient characteristics of matched groups, see Table 1.

For the sensitivity analysis, we iterated the control group matching with a regression model that included disease severity, cirrhosis etiology and proton pump inhibitor use. Sixty percent of patients from the initial control cohort (only matched for liver disease severity) were replaced during this process. In addition, this control group was well comparable to the LOLA group in regard to age (*p* = 0.5), sex (*p* = 0.4), etiology (*p* > 0.99), Child-Pugh score (*p* = 0.3), MELD score (*p* = 0.9), PPI use (*p* > 0.99) and lactulose use (*p* > 0.99). None of the patients in the control group were taking non-absorbable antibiotics. Patient characteristics are compared in Table 2.

### 3.2. Association of LOLA Intake with Microbiome Composition

The analysis of the microbiome compositions showed no significant differences in alpha diversity (Chao1: *p* = 0.4, Evenness: *p* = 0.2; Richness: *p* = 0.8, Shannon: *p* = 0.2) or beta diversity (ANOSIM: R = −0.045, *p* = 0.9; RDA: Variance = 27.46, F = 0.98, *p* = 0.6, NMDS: stress = 0.235), and ANCOM could not identify any significant differences between the groups. LEfSe analysis identified six genera to be differentially abundant between groups: *Flavonifractor* and *Oscillospira* was associated with LOLA therapy, *Ruminococcaceae_UCG003*, *Butyricimonas*, *Desulfovibrio* and an uncultured, not further identified bacterium were associated with controls without LOLA therapy (see also Figure 2). This result was also observed when patients with intake of lactulose or PPI or rifaximin were excluded from the analysis (see Appendix A).

Similar results were obtained in the sensitivity analysis using the control group matched for liver disease severity, etiology and PPI use. Alpha-diversity (Chao1: *p* = 0.7, Evenness: *p* = 0.9; Richness: *p* = 0.6, Shannon: *p* = 0.8) and beta-diversity (ANOSIM: R = −0.006, *p* = 0.5; RDA: Variance = 28.29, F = 1.04, *p* = 0.2, NMDS: stress = 0.224) did not show significant differences between patients with and without LOLA use, consistent with the previous analysis. LEfSe identified five differentially abundant genera; *Parabacteriodes*, *Flavonifractor* and *Oscillospira* were more abundant in patients treated with LOLA, *Subdoligranulum* and *Anaerostipes* were more abundant in patients not taking LOLA (see also Appendix A). ANCOM could identify the genus *Flavonifractor* to be differentially abundant between groups.

The genera *Flavonifractor* and *Oscillospira* re-emerged in the sensitivity analysis using the control group matched for liver disease severity, etiology and PPI use, and the differences in *Flavonifractor* abundance could be reproduced by ANCOM. Both taxa are more abundant in patients treated with LOLA compared to patients not taking LOLA. *Flavonifractor* was 4.9-fold and 10.6-fold higher and *Oscillospira* was 14.6-fold and 3.4-fold higher in patients with LOLA intake compared to patients without in the initial analysis and in the sensitivity analysis using the control group matched for liver disease severity, etiology and PPI use, respectively, as shown in Appendix A.

### 3.3. Metabolomic Analysis of Urine, Stool and Serum Samples

To identify metabolites discriminating patients treated with LOLA from patients not taking LOLA, we used PCA and ROC analysis of metabolite panels obtained for urine, stool and serum samples using NMR spectroscopy. Whereas PCA showed no clear separation of the groups (PCA plots are given in Appendix A), ROC analysis was able to identify metabolite (ratios)-discriminating patients treated with LOLA from patients not taking LOLA.

#### 3.3.1. Urine Metabolome

Urine metabolomics data of 29 patients were available (14 LOLA, 15 noLOLA). In total, 41 metabolites and the 20 most distinctive metabolite ratios were analyzed. Of these, 14 biomarkers showed a c value above 0.7 in the AUROC analysis and significant differences between groups (uncorrected *t* test). Because of the considerable risk of overfitting, the analysis was repeated in the sensitivity analysis dataset using the control group matched for liver disease severity, etiology and PPI use to select the more robust biomarkers. This dataset consisted of 27 patients (14 LOLA, 13 noLOLA). Biomarker analysis identified 13 potential biomarkers to distinguish between patients with and without LOLA use. The overlap in potential biomarkers between the initial analysis and the sensitivity analysis included only one biomarker: ethanol to acetic acid ratio was higher in the noLOLA group. Neither ethanol nor acetic acid showed considerable predictive power as a standalone biomarker. Details are given in Table 3. All urine samples contained at least minimal concentrations of ethanol, although more than half of the patients reported not to drink alcohol at all. Patient-reported alcohol consumption did not correlate with urine ethanol concentrations (r_s_ = −0.057; *p* = 0.8) nor with urine ethanol to acetic acid ratio (r_s_ = −0.051; *p* = 0.9). Moreover, urine ethanol concentrations were also not correlated with serum ethanol concentrations (r_s_ = −0.035; *p* = 0.9). Patient-reported alcohol consumption was comparable between groups.

#### 3.3.2. Stool Metabolome

Valid stool metabolomics data were available in 27 of the 30 patients (13 LOLA, 14 noLOLA). In total, 48 metabolites and the 20 most distinctive metabolite ratios were analyzed. Biomarker identification showed 19 potential biomarkers in the initial analysis matched for liver disease severity and 12 in the sensitivity analysis matched for liver disease severity, etiology and PPI use, 3 of which could be validated in both datasets: propylene glycol to isopropyl alcohol ratio, propylene glycol to valeric acid ratio, and valeric acid to glycerol ratio. Neither propylene glycol, isopropyl alcohol, valeric acid nor glycerol showed significant predictive potential as standalone biomarkers. Of note, although valeric acid to glycerol ratio showed significant group differences and c values above 0.7 in both cohorts, the predictive power assessed by AUROC did not reach statistical significance in the sensitivity analysis. Details are given in Table 4.

#### 3.3.3. Serum Metabolome

Serum metabolome datasets with 42 compounds of 27 patients were available. Biomarker identification selected several biomarkers that indicated hepatic encephalopathy and LOLA adherence. Biomarkers that showed significant predictive power in the initial analysis and the sensitivity analysis are listed in Table 5. As expected, the presence of ornithine in the serum of LOLA patients dominates the biomarker selection and thereby validates drug adherence and absorption of oral LOLA.

### 3.4. Serum- and Fecal-Targeted Metabolomics and Clinical Characteristics

Liver parameters ALT, AST, AP, GGT, albumin as well as total protein, bilirubin, and PZINR, were well balanced between the groups. Patients with LOLA intake showed significantly lower fecal zonulin levels, decreased insulin-like growth factor and slower gait speed compared to patients without LOLA intake. All other tested biomarkers did not show significant differences. Details are given in Table 6.

In the sensitivity analysis, none of the above-described differences could be reproduced. Only the reduced IGF-1 levels showed a similar but not statistically significant difference between groups. Details are given in Appendix A. The decreased IGF-1 levels might be associated with the higher rate of sarcopenic patients in the LOLA group. Although the difference was not statistically significant, there was a slight overrepresentation of sarcopenic patients in the LOLA group. Of the fifteen patients in the group, nine were sarcopenic, three pre-sarcopenia and three non-sarcopenic. In the control group, however, only six patients were sarcopenic, three pre-sarcopenic and six non-sarcopenic. We therefore matched a third control dataset to the LOLA patients based on sarcopenia diagnosis only for this analysis (irrespective of liver disease) that still showed significantly higher levels of IGF-1 compared to the LOLA group (93.1 ± 51.1 versus 48.3 ± 28.1 ng/mL, *p* = 0.01), indicating the close connection of hepatic encephalopathy to muscle metabolism in cirrhosis.

## 4. Discussion

Our retrospective analysis of 15 patients with oral LOLA intake for at least one month and propensity-score-matched controls showed that patients with LOLA intake show a higher abundance of the genera *Flavonifractor* and *Oscillospira* in the intestinal microbiome, a reduced ratio of ethanol to acetic acid in urine, an increased ratio of propylene glycol to isopropyl alcohol in stool, increased ratio of propylene glycol to valeric acid in stool, decreased ratio of valeric acid to glycerol in stool, increased levels of ornithine and decreased levels of leucine and isoleucine as well as lower levels of IGF-1 in serum.

*Flavonifractor* and *Oscillospira* are rather closely related bacterial taxa in the human intestinal microbiome. According to the Silva database, both genera are classified as Firmicutes (phylum), Clostridia (Class), Oscillospirales (order), Oscillospiraceae (family). *Flavonifractor* that can be induced by green tea consumption and can exert anti-inflammatory properties in murine DSS colitis models [25]. In humans, the role of the genus *Flavonifractor* is still undetermined. Its abundance was previously associated with the consumption of Mediterranean diet [26], which is generally regarded as beneficial for human health. In non-alcoholic liver disease, *Flavonifractor* consistently showed low abundance compared to healthy controls [27]. In addition, according to the GMrepo database, *Flavonifractor* is more abundant in healthy volunteers than in cirrhosis, non-alcoholic fatty liver disease, obesity/adiposity, and diabetes mellitus (topic-related selection) [28]. Conversely, *Flavonifractor* has been associated with the microbiome of colorectal cancer patients in India [29]. *Oscillospira* has previously been associated with low BMI, constipation, longer sleep time, higher HDL levels, lower uric acid and triglyceride levels [30,31]. In a previous study, we showed that lower HDL levels are predictive for the development of complications in patients with compensated cirrhosis and predictive for survival in cirrhotic patients with acute decompensation [32,33]. Taken together, the alterations in the microbiome of LOLA patients could generally be regarded as beneficial for patients with liver disease. However, due to the retrospective design and the low sample size, confounding factors cannot be ruled out, e.g., although not statistically significant, patients in the LOLA group more often also received rifaximin.

In the urine metabolome, patients in the LOLA group showed lower ethanol to acetic acid ratio. The ethanol to acetic acid ratio might be an indicator for the ethanol elimination process. When alcohol is consumed, it is converted to acetaldehyde—a toxic intermediate—and then quickly reduced to acetate. A reduced ratio of ethanol to acetic acid might indicate a more advanced state in the ethanol elimination process, either because of an earlier consumption time or accelerated conversion. Both patient groups reported similar amounts of alcohol intake and did not show any difference in serum or urine alcohol levels. An earlier starting point would therefore necessitate a higher level of intoxication in the LOLA group, which was not reported by the patients. However, in patients with present or past alcohol use disorder, alcohol consumption is notoriously hard to approximate, since some patients tend to embellish their consumption [34]. Appropriately powered and designed studies are necessary to properly explain the reduced ethanol to acetic acid ratio in the urine metabolome of LOLA patients and to further explore the possible effects of LOLA on the ethanol elimination rate in cirrhotic patients. In the stool microbiome, the most prominent biomarker was the propylene glycol to isopropyl alcohol ratio. Propylene glycol is naturally occurring in mushrooms and sesame seed but is also ubiquitously used in the cosmetic, food and drug industries. Isopropyl alcohol is a common substance in various nutritious items, including apples and onions, which are a crucial part of the local cuisine of the patients´ catchment area. The relevance in health and disease is undetermined at the moment. An increase in this ratio in LOLA patients might indicate an increase in therapeutics and decrease in fresh food consumption. However, this exploratory study is not designed to answer this question. Valeric acid is a bacterial metabolite, among others produced by *Flavonifractor plautii* as a product of flavonoid degradation [35]. This short chain fatty acid can also be produced by odd-chain elongation of ethanol and propionate or by hydrolysis of valerate esters used as fruity tasting food additives [36]. We observed low levels of valeric acid to glycerol ratio in the stool metabolome of LOLA patients, which is consistent with previous observations of low valerate levels in patients with recurring hepatic encephalopathy and might therefore reflect the disease rather than a therapy-related change in the metabolome [37]. Similarly, serum marker reflect a typical decrease in branched chained amino acids in patients with hepatic encephalopathy [38]. The lack of isoleucine probably would have been even more pronounced if the patients were not treated with LOLA, since LOLA supposedly increases isoleucine levels in serum [39]. Moreover, the increased levels of ornithine confirm therapy adherence of the patients and adequate reabsorption of oral LOLA in these patients [40].

IGF-1 was consistently lower in patients with LOLA, although we corrected for liver disease and sarcopenia in two separate matched cohort. In this pilot study, we could not discern the reason for this decrease nor could we find probable mechanistic indicators. It is likely that the pathophysiological changes in patients with hepatic encephalopathy, especially the high levels of ammonia, negatively influence muscle metabolism as it was previously shown in pufferfish and suggested in cirrhosis and hepatic encephalopathy [41,42].

In conclusion, patients with LOLA therapy for at least one month showed a potentially beneficial increase in *Flavonifractor* and *Oscillospira* in their microbiome compared to patients without LOLA therapy. Changes in the stool and serum metabolome mainly reflected pathophysiological changes of hepatic encephalopathy and identified possible therapeutic targets, such as valeric acid that is reduced in patients with hepatic encephalopathy but whose microbial production might be elevated by LOLA-associated changes in *Flavonifractor* abundance. Lastly, low levels of IGF-1 demonstrated a link between hepatic encephalopathy and muscle metabolism, a potential new avenue for LOLA use in liver cirrhotic patients.

## Figures and Tables

**Figure 1 nutrients-14-00748-f001:**
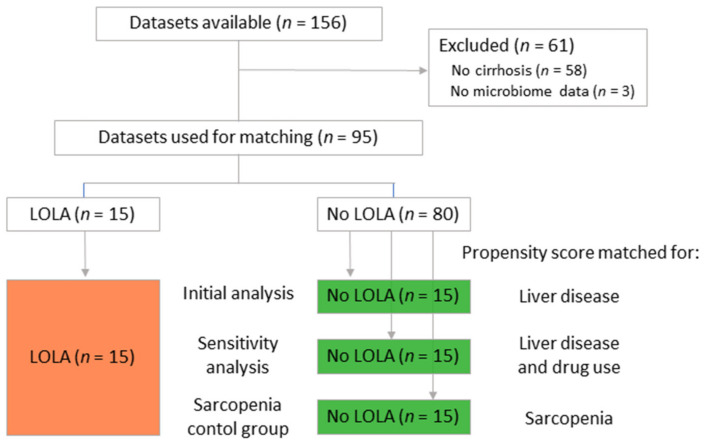
Flow diagram for patient selection. Analyzed groups are given in colored boxes.

**Figure 2 nutrients-14-00748-f002:**
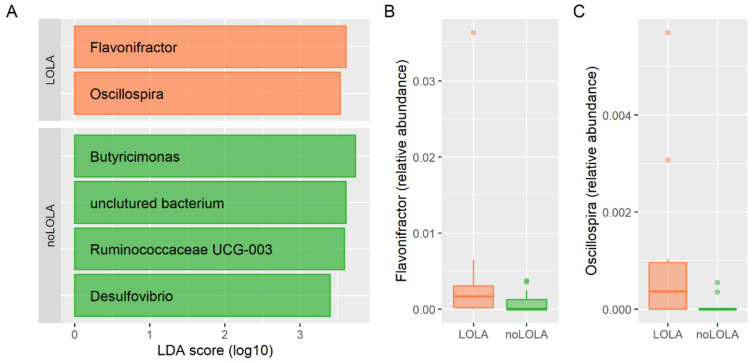
(**A**) Genera associated with LOLA therapy (orange bars—LOLA) or controls (green bars—noLOLA) determined by LDA effect size (LEfSe). (**B**,**C**) Relative abundances of genera *Flavonifractor* and *Oscillospira* in patient with and without LOLA intake.

**Table 1 nutrients-14-00748-t001:** Characteristics of LOLA-receiving patients and controls matched for liver disease severity.

Parameter	LOLA (*n* = 15)	noLOLA (*n* = 15)	*p* Value
Age	63 (6.5)	61 (15.1)	0.9
Sex (m/f)	13/2 (87/13%)	11/4 (73/27%)	0.7
Etiology (Alc/Non-alc)	10/5 (67/33%)	9/6 (60/40%)	0.7
Child-Pugh score	7 (1.5) *	7 (1.5) *	>0.99
MELD score	12.8 (3.3)	11.9 (2.7)	0.5
PPI use/non-use	7/8 (47/53%)	6/9 (40/60%)	>0.99
Lactulose use/non-use	5/10 (33/67%)	3/12 (20/80%)	0.7
Non-absorbable antibiotics use/non-use	4/11 (36/64%)	0/15 (0/100%)	0.1
Sarcopenia diagnosis (no/pre-/sarcopenia)	3/3/9 (20/20/60%)	6/3/6 (40/20/40%)	0.4

Values are given as mean (standard deviation) or count (%), unless otherwise stated; * values are given as median (interquartile range); LOLA: patients with LOLA intake; noLOLA: patients without LOLA intake; Alc: alcoholic cirrhosis; MELD: model of end stage liver disease; PPI: proton pump inhibitor.

**Table 2 nutrients-14-00748-t002:** Patient characteristics of LOLA-receiving patients and controls matched for liver disease severity, etiology and PPI use.

	LOLA (*n* = 15)	noLOLA (*n* = 15)	*p* Value
Age (years)	63 (6.5)	62 (7.7)	0.5
Gender (m/f)	13/2 (87/13%)	10/5 (67/33%)	0.4
Etiology (Alc/Non-alc)	10/5 (67/33%)	10/5 (67/33%)	>0.99
Child-Pugh score	7 (1.5) *	7 (2.5) *	0.3
MELD score	12.8 (3.3)	12.7 (4.3)	0.9
PPI use/non-use	7/8 (47/53%)	7/8 (47/53%)	>0.99
Lactulose use/non-use	5/10 (33/67%)	4/11 (36/64%)	>0.99
Non-absorbable antibiotics use/non-use	4/11 (36/64%)	0/15 (0/100%)	0.1
Sarcopenia diagnosis (no/pre-/sarcopenia)	3/3/9 (20/20/60%)	7/3/5 (47/20/33%)	0.3

Values are given as mean (standard deviation) or count (%), unless otherwise stated; * values are given as median (interquartile range); LOLA: patients with LOLA intake; noLOLA: patients without LOLA intake; Alc: alcoholic cirrhosis; MELD: model of end stage liver disease; PPI: proton pump inhibitor.

**Table 3 nutrients-14-00748-t003:** Predictive power of urine ethanol to acetic acid ratio and its components for patients with and without LOLA intake in both the initial analysis (groups matched for liver disease severity) and the sensitivity analysis (groups matched for liver disease severity, etiology and PPI use).

	Initial Analysis			Sensitivity Analysis		
Biomarker	LOLA	noLOLA	AUROC (95%CI)	LOLA	noLOLA	AUROC (95%CI)
**Ethanol to acetic acid ratio**	**−0.59 (0.85)**	**0.30 (0.51)**	**0.83 (0.61–0.95)**	**−0.46 (0.78)**	**0.07 (0.64)**	**0.76 (0.54–0.92)**
Ethanol	−0.11 (0.60)	0.00 (0.63)	0.60 (0.37–0.80)	−0.02 (0.54)	−0.16 (0.84)	0.53 (0.31–0.78)
Acetic acid	0.42 (0.91)	−0.27 (0.79)	0.68 (0.46–0.87)	0.42 (0.87)	−0.24 (0.8)	0.69 (0.78–0.88)

Values are given as mean (standard deviation) or AUROC c value (95%CI); LOLA: patients with LOLA intake; noLOLA: patients without LOLA intake; AUROC: area under the receiver operator curve; 95%CI: 95% confidence interval; biomarkers with significant predictive power are printed in bold.

**Table 4 nutrients-14-00748-t004:** Predictive power of stool metabolite ratios that distinguish between patients with and without LOLA intake in both the initial analysis (matched for liver disease severity) and the sensitivity analysis (matched for liver disease severity, etiology and PPI use).

	Initial Analysis			Sensitivity Analysis		
Biomarker/Ratio	LOLA	noLOLA	AUROC (95%CI)	LOLA	noLOLA	AUROC (95%CI)
**Propylene glycerol to isopropyl alcohol**	**0.54 (0.77)**	**−0.62 (0.93)**	**0.84 (0.65–0.96)**	**0.46 (0.77)**	**−0.67 (1.00)**	**0.78 (0.56–0.93)**
**Propylene glycerol to valeric acid**	**0.59 (1.01)**	**−0.40 (0.86)**	**0.74 (0.52–0.89)**	**0.57 (0.91)**	**−0.28 (0.85)**	**0.76 (0.53–0.90)**
**Valeric acid to glycerol**	**−0.23 (1.00)**	**0.41 (0.78)**	**0.73 (0.51–0.90)**	−0.19 (0.98)	0.12 (0.63)	0.72 (0.49–0.89)
Propylene glycerol	0.30 (1.30)	−0.08 (0.70)	0.65 (0.42–0.82)	0.31 (0.98)	−0.25 (1.02)	0.65 (0.40–0.84)
Isopropyl alcohol	−0.21 (1.04)	0.44 (0.83)	0.69 (0.47–0.87)	−0.06 (0.88)	0.24 (1.22)	0.62 (0.38–0.82)
Valeric acid	−0.27 (1.28)	0.32 (0.74)	0.36 (0.19–0.58)	−0.13 (0.87)	−0.03 (1.21)	0.67 (0.47–0.88)
Glycerol	0.07 (1.09)	−0.24 (0.85)	0.62 (0.42–0.82)	0.08 (0.97)	−0.15 (1.02)	0.61 (0.39–0.81)

Values are given as mean (standard deviation) or AUROC c value (95%CI); LOLA: patients with LOLA intake; noLOLA: patients without LOLA intake; AUROC: area under the receiver operator characteristics curve; 95%CI: 95% confidence interval; biomarkers with significant predictive power are printed in bold.

**Table 5 nutrients-14-00748-t005:** Predictive power of serum metabolite ratios that distinguish between patients with and without LOLA intake in both the initial analysis (groups matched for liver disease severity) and the sensitivity analysis (groups matched for liver disease severity, etiology and PPI use).

	Initial Analysis			Sensitivity Analysis		
Biomarkers/Ratios	LOLA	noLOLA	AUROC (95%CI)	LOLA	noLOLA	AUROC (95%CI)
**Ornithine**	**0.53 (1.06)**	**–0.42 (0.85)**	**0.76 (0.57–0.93)**	**0.54 (1.04)**	**–0.38 (0.87)**	**0.78 (0.59–0.93)**
**Isoleucine**	**−0.54 (1.23)**	**0.37 (0.59)**	**0.79 (0.57–0.93)**	**–0.51 (0.98)**	**0.46 (0.88)**	**0.76 (0.54–0.92)**
**Leucine**	**–0.37 (1.24)**	**0.36 (0.57)**	**0.74 (0.52–0.92)**	**–0.35 (1.07)**	**0.54 (0.7)**	**0.73 (0.50–0.90)**
**Isoleucine to ornithine**	**–0.7 (0.9)**	**0.52 (0.77)**	**0.88 (0.70–0.97)**	**–0.8 (0.94)**	**0.62 (0.62)**	**0.91 (0.77–1.00)**
**Acetone to ornithine**	**–0.58 (0.97)**	**0.51 (0.83)**	**0.83 (0.63–0.95)**	**–0.52 (0.99)**	**0.37 (0.76)**	**0.75 (0.54–0.92)**
**Lysine to ornithine**	**–0.53 (1.04)**	**0.46 (0.71)**	**0.82 (0.63–0.96)**	**–0.54 (1.10)**	**0.53 (0.72)**	**0.82 (0.64–0.98)**
**Leucine to ornithine**	**–0.55 (1.16)**	**0.46 (0.61)**	**0.84 (0.67–0.90)**	**–0.58 (1.16)**	**0.56 (0.64)**	**0.87 (0.70–0.98)**
**Ethanol to ornithine**	**–0.54 (0.95)**	**0.49 (0.87)**	**0.81 (0.61–0.95)**	**–0.52 (1.03)**	**0.46 (0.84)**	**0.80 (0.60–0.95)**
**Ornithine to threonine**	**0.55 (1.05)**	**–0.52 (0.77)**	**0.83 (0.66–0.96)**	**0.64 (0.93)**	**–0.49 (0.72)**	**0.90 (0.75–0.98)**
**Aspartic acid to ornithine**	**–0.62 (1.1)**	**0.46 (0.71)**	**0.82 (0.62–0.97)**	**–0.6 (1.07)**	**0.38 (0.60)**	**0.79 (0.58–0.93)**
**Ornithine to serine**	**0.56 (1.08)**	**–0.5 (0.74)**	**0.82 (0.61–0.95)**	**0.59 (0.95)**	**–0.34 (0.63)**	**0.86 (0.65–0.97)**
**Valine to ornithine**	**–0.57 (1.14)**	**0.41 (0.66)**	**0.78 (0.60–0.93)**	**–0.62 (1.19)**	**0.52 (0.59)**	**0.85 (0.66–0.99)**
**Arginine to ornithine**	**–0.49 (1.07)**	**0.41 (0.73)**	**0.78 (0.57–0.92)**	**–0.52 (1.06)**	**0.37 (0.63)**	**0.82 (0.64–0.97)**
Acetone	–0.27 (0.75)	0.37 (1.05)	0.70 (0.46–0.88)	0.05 (0.74)	0.01 (1.00)	0.54 (0.31–0.75)
Lysine	–0.18 (0.63)	0.25 (0.90)	0.72 (0.48–0.90)	–0.07 (0.53)	0.38 (0.90)	0.72 (0.50–0.89)
Ethanol	–0.05 (0.85)	0.27 (0.88)	0.61 (0.41–0.81)	0.21 (0.74)	0.13 (1.04)	0.50 (0.29–0.74)
Threonine	–0.07 (1.11)	0.22 (0.92)	0.63 (0.36–0.83)	–0.30 (0.92)	0.28 (0.85)	0.74 (0.51–0.89)
Aspartic acid	–0.29 (0.80)	0.17 (1.19)	0.56 (0.34–0.76)	–0.25 (0.76)	0.09 (1.21)	0.55 (0.34–0.75)
Serine	0.03 (1.12)	0.12 (0.83)	0.46 (0.23–0.68)	–0.17 (0.90)	–0.07 (0.86)	0.62 (0.41–0.82)
Valine	–0.44 (1.23)	0.26 (0.62)	0.70 (0.49–0.90)	–0.40 (1.05)	0.43 (0.92)	0.73 (0.52–0.90)
Arginine	–0.11 (1.02)	0.15 (0.42)	0.56 (0.34–0.78)	–0.15 (0.86)	0.10 (0.88)	0.67 (0.42–0.86)

Values are given as mean (standard deviation) or AUROC c value (95%CI); LOLA: patients with LOLA intake; noLOLA: patients without LOLA intake; AUROC: area under the receiver operator characteristics curve; 95%CI: 95% confidence interval; biomarkers with significant predictive power are printed in bold.

**Table 6 nutrients-14-00748-t006:** Liver disease parameters, gut inflammation and permeability markers, indicators of sarcopenia and neutrophil function in the initial analysis (groups matched for liver disease severity). Values are given in mean (standard deviation). Significant differences are marked in bold print.

Parameter	LOLA (*n* = 15)	noLOLA (*n* = 15)	*p* Value
Alanine aminotransferase (U/L)	38.8 (22.8)	39.5 (15.6)	0.6
Aspartate aminotransferase (U/L)	67.7 (41.9)	66.2 (35)	0.9
Alkaline phosphatase (U/L)	137.7 (59.2)	125.7 (68.8)	0.3
Gamma-glutamyltransferase (U/L)	134.5 (92.8)	121.9 (105.3)	0.6
Albumin (g/dL)	3.2 (0.5)	3.3 (0.5)	0.4
Bilirubin (mg/dL)	2.3 (1.8)	2.4 (1.9)	>0.99
Prothrombin time international normalized ratio	1.4 (0.2)	1.3 (0.2)	0.2
Total protein (g/dL)	6.8 (0.9)	7 (0.9)	0.3
Fecal calprotectin (ng/mL)	101.4 (103.9)	80.3 (64.6)	0.7
**Fecal zonulin (ng/mL)**	**161.2 (219.9)**	**205.2 (203.5)**	**0.036**
Diamine oxidase (U/mL)	24 (11.9)	23.3 (15.3)	0.6
LPS binding protein (µg/mL)	16.7 (7)	20.4 (9.8)	0.4
C-reactive protein (mg/L)	10.3 (13.6)	8.1 (15.2)	0.4
soluble Cluster of Differentiation 14 (µg/mL)	1.8 (0.4)	2 (0.8)	0.7
Fibroblast growth factor (ng/mL)	0.3 (0.6)	0.3 (0.2)	0.3
Irisin (µg/mL)	2 (1.5)	1.9 (1)	0.8
Myostatin (ng/mL)	44 (34.1)	38.6 (15.2)	0.8
**Insulin-like growth factor 1 (ng/mL)**	**48.3 (28.1)**	**75.5 (38.4)**	**0.029**
Chair rise test (s)	25 (15.9)	16.7 (3.8)	0.1
**Gait speed (m/s)**	**0.8 (0.3)**	**1.1 (0.3)**	**0.049**
Midarm muscle circumference (mm)	256.6 (63.7)	260.4 (48.1)	0.7
Hand grip strength (kg)	30.6 (10.6)	28.8 (5.8)	0.5
Body mass index (kg/m^2^)	27.5 (6)	28.5 (5.8)	0.6
Resting burst of neutrophils (% of neutrophils)	2.4 (0.9)	2.6 (1.9)	0.7
Resting burst of neutrophils (GMFI)	174 (102.9)	239.8 (223.3)	0.4
Neutrophil priming (% of neutrophils)	3.2 (1.4)	2.7 (1.2)	0.5
Neutrophil priming (GMFI)	148.7 (58.7)	194.8 (99.1)	0.1
ROS production after *E. coli* stimulation (% of neutrophils)	96.7 (5.2)	96.8 (4.5)	0.6
ROS production after *E. coli* stimulation (GMFI)	1034.2 (547.1)	753.7 (340.1)	0.1

LOLA: patients with LOLA intake; noLOLA: patients without LOLA intake; LPS: lipopolysaccharide; GMFI: geometric mean of fluorescence intensity.

## Data Availability

The data presented in this study are available on request from the corresponding author. The data are not publicly available yet due to the ongoing collection of data in this study.

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
