# Peer review of "Oral Intake of L-Ornithine-L-Aspartate Is Associated with Distinct Microbiome and Metabolome Changes in Cirrhosis"

_nutrients, 2022, doi:10.3390/nu14040748_

Round 1
Reviewer 1 Report
In this manuscript authors have described the role of L-Ornithine L-Aspartate (LOLA) in inducing beneficial microbiome and metabolomic changes in liver cirrhosis patients. Authors have found that in LOLA treated group Flavonifractor and Oscillospira were more abundant compared to control group. Authors have also found that in urine the ratio of ethanol to acetic acid was lower in LOLA treated group. In addition to this serum IGF1 levels were less in LOLA treated group. I found this study to be interesting and well executed study. But it had a very few human subjects and might need bigger sample size.
Major comments:
- Sample size included in the analysis is small. But author have used stricter measures for including subjects in the group.
- I think it is important to validate the 16rDNA seq findings using qPCR probes specific for Flavonifractor and Oscillospira in the LOLA treated and no LOLA treated groups. Entire study hinges on this finding so having another method for verifying this result is critical.
- LOLA treated group 4 subjects used laxatives/rifaximin/proton pump inhibitor. Laxatives can cause long term effects on microbiome (https://www.cell.com/cell/fulltext/S0092-8674(18)30585-3?_returnURL=https%3A%2F%2Flinkinghub.elsevier.com%2Fretrieve%2Fpii%2FS0092867418305853%3Fshowall%3Dtrue)
Reanalyze the 16sDNA seq data in the LOLA treated group and compare it with other 11 subjects that did not take laxatives during the study and show that Flavonifractor and Oscillospira are still high in the LOLA treated group in the 11 subjects. It excludes the possibility of confounding effects caused by laxatives or other antibiotic use.
- LOLA treatment had any beneficial effects on the cirrhotic patients in this study?
Minor comments:
- Authors might need to fill out details in following sections a) author contributions b) informed consent statement c) data availability statement d) acknowledgements e) conflict of interest .
Author Response
We thank the reviewer for their time and their valuable comments
Reviewer 1
We thank the reviewer for his/her valuable suggestions
- Sample size included in the analysis is small. But author have used stricter measures for including subjects in the group.
Thank you for this comment, we aimed for stringent statistical measures to compensate for the small sample size
- I think it is important to validate the 16rDNA seq findings using qPCR probes specific for Flavonifractor and Oscillospira in the LOLA treated and no LOLA treated groups. Entire study hinges on this finding so having another method for verifying this result is critical.
We agree with the reviewer that the verification of sequencing results by qPCR would be ideal. However, our analysis was performed as a post hoc analysis of a cohort study (intended to study the gut-liver-muscle axis NCT03080129), we unfortunately do not have enough biomaterial (stool or isolated DNA) left to perform this analysis. But we aim to perform a prospective study based on these findings, where we will definitely plan to collect enough stool samples to be prepared for such analyses.
- LOLA treated group 4 subjects used laxatives/rifaximin/proton pump inhibitor. Laxatives can cause long term effects on microbiome. Reanalyze the 16sDNA seq data in the LOLA treated group and compare it with other 11 subjects that did not take laxatives during the study and show that Flavonifractor and Oscillospira are still high in the LOLA treated group in the 11 subjects. It excludes the possibility of confounding effects caused by laxatives or other antibiotic use.
We agree with the reviewer. that other drugs can also have an effect on the microbiome composition. We therefore followed the reviewer’s suggestion and excluded the patients taking laxatives (i.e. lactulose) for a subgroup analysis. This analysis showed that Flavonifractor and Oscillospira are still higher in the LOLA group than in the control groups. For details see the following figure (A-B). Similar subgroup analysis was done for PPI (C-D) and rifaximin (E-F) with similar results. Due to the low power and robustness of these results, we would prefer not to include them in the manuscript but if the reviewer thinks that this is necessary we could include the analysis in the supplementary file.
Regarding antibiotic use: The only antibiotic allowed in the study was the non-absorbable antibiotic rifaximin. Use of other antibiotics was part of the exclusion criteria.
Figure 1R: relative abundance of Oscillospira and Flavonifractor in patients with and without LOLA use and no concomitant lactulose use (A-B), no concomitant PPI use (C-D) and no concomitant rifaximin use (E-F).
- LOLA treatment had any beneficial effects on the cirrhotic patients in this study?
The study that was the basis for our analysis, is designed as a cross sectional study, therefore we do not have any outcome data on the patients from this study. But due to our results, we are now planning a prospective study to assess the effect of LOLA on gut microbiome in cirrhosis and in this study we definitely will collect outcome data.
Minor comments:
- Authors might need to fill out details in following sections a) author contributions b) informed consent statement c) data availability statement d) acknowledgements e) conflict of interest .
This information can be found from line 327 onwards (before the references section). We apologize for the inconvenience that this information was not disclosed to the reviewer.

Reviewer 2 Report
The manuscript by Horvath et al comprehensively examines the effects of LOLA in a small cohort of cirrhotic patients. Changes in microbiome, metabolome and clinical markers are examined following LOLA treatment, demonstrating some minor effects with potential clinical impact. This is a well written manuscript with some interesting results that likely warrant a larger cohort with improved statistical power. The data is well presented and conclusions are appropriate, however there is some information lacking that would improve reader comprehension. My suggestions are as follows: 1. A short explanation of the neurotoxic effects of ammonia should be included in the intro regarding its HE relevance 2. Significant information regarding the neutrophil function tests are lacking. How were resting burst, E coli cocultures etc performed? Please elaborate 3. The inclusion of three different control groups is quite confusing, especially with regard to your results section. It is quite difficult to assess which group is used for the different analyses. I would suggest elaborating in the methods section as to which controls group is used for which analysis. It may be advantageous as well to include sarcopenia data in Tables 1 and 2. 4. How long were patients treated with LOLA? Yes, they were treated for >1 month but what is the range? Any indication that longer treatment duration improves efficacy? 5. Please include data on patient lactulose tx. I imagine this is included in the laxative data, however it should be considered separately due to its effects on HE. 6. May rifaximin have affected the LOLA group? Although there is no statistical difference between groups, this is a potentially meaningful difference (36% vs 0%). This should be touched on in the discussion Minor sp/grammar errors (Minor spell check required) - Intro- “Since oral LOLA gets into contact” should be “comes into contact” - Table 6- “insulin-line growth factor” - Discussion- IFG1 should be IGF1, “data base” ïƒ database - Supplement- “soluble Cluster of Differentiation (µg/mL)” Which one? CD14??
Author Response
Reviewer 2
We thank the reviewer for his/her valuable suggestions
- A short explanation of the neurotoxic effects of ammonia should be included in the intro regarding its HE relevance
Thank you for pointing this out. This information was added.
- Significant information regarding the neutrophil function tests are lacking. How were resting burst, E coli cocultures etc performed? Please elaborate
We apologize for not including this information. It is now included into the supplementary methods.
- The inclusion of three different control groups is quite confusing, especially with regard to your results section. It is quite difficult to assess which group is used for the different analyses. I would suggest elaborating in the methods section as to which controls group is used for which analysis. It may be advantageous as well to include sarcopenia data in Tables 1 and 2.
We are sorry for being not precise enough, we revised the methods and results including Figure 1 accordingly to improve clarity. Data on sarcopenia diagnosis was added to the tables
- How long were patients treated with LOLA? Yes, they were treated for >1 month but what is the range? Any indication that longer treatment duration improves efficacy?
Information on the range of LOLA treatment was added. To investigate if longer LOLA treatment is associated with more pronounced changes in the microbiome we ran Spearman correlation analysis with the abundances of Flavonifractor/Oscillospira and the approximate duration of intake. This analysis showed that there is no correlation between the duration of intake and the abundance of Flavonifractor (rs=-0.17, p=0.5) or the abundance of Oscillospira (rs=0.29, p=0.3). We have to point out, that due to the retrospective collection of these data, the accuracy in collecting information on the duration of drug intake may not be as good as in prospective settings.
- Please include data on patient lactulose tx. I imagine this is included in the laxative data, however it should be considered separately due to its effects on HE.
In all cases, lactulose was the laxative used, we therefore amended the wording to “lactulose” instead of “laxative”
- May rifaximin have affected the LOLA group? Although there is no statistical difference between groups, this is a potentially meaningful difference (36% vs 0%). This should be touched on in the discussion
We added this consideration to the discussion. Please also see the answer to comment 3 of reviewer 1 where we performed a subgroup analysis and showed (albeit with a low power and robustness due to sample size) that the result regarding Flavonifractor/Oscillospira is not changed when rifaximin use is excluded.
Minor sp/grammar errors (Minor spell check required) - Intro- “Since oral LOLA gets into contact” should be “comes into contact” - Table 6- “insulin-line growth factor” - Discussion- IFG1 should be IGF1, “data base” à database - Supplement- “soluble Cluster of Differentiation (µg/mL)” ßWhich one? CD14??
Thank you for the careful review, we corrected all typos and recheck spelling throughout the whole manuscript carefully

Round 2
Reviewer 1 Report
According to the suggestions of the reviewers, authors have performed reanalysis to exclude the possibility of confounding effects caused by the use of antibiotics on the LOLA treated subjects (Figure 1R). I request authors to include it in the supplementary figures and mention it in the text.
Author Response
Dear Reviewer
As requested, we added the graphs to the supplements and mention the graph in the main text.
Sincerely
Vanessa Stadlbauer